behaviour, ecology, physiology

animal personality, temperament, physiological profiles, behavioural consistency, behavioural syndromes, fitness

**Author for correspondence:**
Sonia A. Cavigelli
e-mail: sac34@psu.edu

# A physiological profile approach to animal temperament: How to understand the functional significance of individual differences in behaviour

Elyse K. McMahon[1,2], Elizabeth Youatt[3] and Sonia A. Cavigelli[2]

[1]Ecology Graduate Program, Huck Institutes of the Life Sciences, Pennsylvania State University, 101 Life Sciences Building, University Park, PA 16802, USA
[2]Biobehavioral Health Department, 219 Biobehavioral Health Building, University Park, PA 16802, USA
[3]Psychology Department, Pennsylvania State University, 140 Moore Building, University Park, PA 16802, USA

EKM, 0000-0001-9022-8022; SAC, 0000-0002-6344-9759

Animal behaviour research has experienced a renewed interest in consistent individual differences (i.e. animal personality or temperament). Recent ecological studies have identified environmental conditions that give rise to the development and evolution of temperaments and to fitness-related outcomes of temperament. Additional literature has also described relationships between temperaments and physiological regulation. However, one-to-one relationships between one behavioural trait and one physiological system do not account for co-selection of behavioural and physiological traits, nor the complex signalling among physiological systems. In the current paper, we review the literature on multiple physiological processes associated with temperament, propose temperament-specific physiological profiles, and focus on next steps to understand the functional significance, evolution and maintenance of temperaments. We propose that to understand causes and consequences of temperament we need to characterize integrative physiological profiles associated with different temperaments.

## 1. Introduction

In the past 20 years, the field of animal behaviour has experienced a renewed interest in individual differences, with a recent focus on 'animal personalities' or 'temperaments'. This resurgence was spurred on by recognition that individually distinct and consistent behavioural traits are not unique to humans, but are widespread across the animal kingdom [1–4]. Recent ecological studies have identified environmental conditions that give rise to the development and evolution of temperaments as well as important fitness-related outcomes of these phenotypes [5–9]. In the current review, we focus on next steps to understand the functional significance, evolution and maintenance of temperament. We propose that it is essential to characterize complex, underlying physiological profiles of temperament in order to understand associated causes and consequences. Doing so will provide a more nuanced, complex and mechanistic understanding of how temperaments evolve and why certain temperaments thrive better in one environment versus another. This information is critical to advance evolutionary and ecological theory, and also applied conservation efforts.

We begin with a brief background on temperament and discuss evidence and limitations related to temperament stability, covariance of temperaments (i.e. behavioural syndromes) and the relationship between temperament and fitness. We review physiological processes that have been associated with temperaments and propose a multi-system physiological framework to incorporate into future studies. This multi-systems physiological approach is

key to understand proximate mechanisms that promote or limit behavioural rigidity/flexibility, covariance and fitness.

## 2. Overview of animal temperament

Many terms are used to refer to reliable or consistent individual differences in behaviour. Recent literature tends to refer to 'personality' or 'behavioural syndromes' which are not synonymous. 'Personality' is often used to indicate consistent behavioural traits within an individual over time and across contexts (although this consistency is not always verified [10], whereas 'syndrome' is used to refer to covarying behavioural responses within an individual (e.g. high aggression associated with elevated boldness behaviour). There has been controversy over the use of 'personality', partially because the definition is often loosely applied, and not necessarily in synchrony with the psychological literature from where the term originates, and because it can be teleological (e.g. [11]). Earlier literature often referred to consistent, individual differences in behaviour as 'temperament', 'alternative behavioural strategies' or 'behavioural phenotypes'. In this review, we focus on basic individual differences in behaviour that are not specific to aspects of life history (e.g. not 'alternative reproductive/mating strategies'), but rather behavioural traits that suggest innate and consistent differences in how individuals respond to all environmental conditions (i.e. 'temperament'). We use the terms 'behavioural trait' or 'temperament' and use Reale et al.'s [12] temperament categories. Additionally, we use a strategy that has been used in human temperament research: a focus on standard temperament categories to identify physiological mechanisms that can impact health and interactions with the environment [13–17]. First, we briefly review three aspects of temperament that may be better understood with more comprehensive physiological information: within-individual temperament consistency, behavioural syndromes and fitness/health outcomes associated with temperament.

### (a) Temperaments are individually consistent

Early animal research included basic observations of behavioural variation among individuals [18,19]. In the late 1990s, it was hypothesized that the 'Big-Five model' of human personality could be used to characterize individual differences in animal behaviour [2,20], and most recently, Réale et al. [12] identified five behavioural dimensions that are frequently studied across species (activity, exploration, boldness, sociability and aggression). Important in the study of temperament, recent work has focused on whether temperaments represent stable traits, as had been documented in humans (e.g. [21,22]). In psychology, multiple behaviour tests or strategies are used to determine the relative consistency of individual temperament. This is not always the case in animal studies because of limited time, resources and/or feasibility in field settings [23]. Additionally, there is no congruence on what defines consistency: is it over time, across conditions or both? However, when repeat testing is conducted with animals, results indicate behavioural consistency across time and across conditions that are in the same order as that for human personality traits (i.e. correlation coefficients of 0.2–0.7 [13]). These differences are thought to reflect systematic differences among individuals that are trait-like. These consistent behavioural traits may reflect consistent physiological underpinnings. By understanding the physiological profile of temperaments, and their relative flexibility, we can better understand biological mechanisms that allow a behavioural trait to persist or to be flexible within an individual.

### (b) Covariance of temperaments

Recent work shows that some behavioural traits covary, which is often referred to as 'behavioural syndromes' or 'behavioural profiles' [24,25]. These behavioural syndromes are not necessarily the same across species or environments. For example, exploration and boldness have been shown to be positively correlated in some species in certain environments, but not in others; the same is true for exploration and sociability [26–31]. On the other hand, across several species, boldness and aggression have been shown to covary (funnel-web spiders [32], crabs [33], sticklebacks [34]; song sparrows [35]). Covarying traits have been characterized as 'coping strategies' (reactive versus proactive) that include inter-related behavioural and physiological traits [36]. For example, proactive individuals in laboratory-based studies are aggressive, bold and behaviourally inflexible, with elevated sympathetic arousal and dampened hypothalamic–pituitary–adrenal (HPA) re/activity [12,24,36–41], whereas reactive individuals show the opposite suite of traits. It has been proposed that covarying behavioural traits emerge as a result of similar physiological processes that underlie different behavioural traits (e.g. elevated sympathetic reactivity promotes both aggression and activity [24]). Thus, a better understanding of the physiological processes that underlie these temperaments can provide the information necessary to determine why certain behaviour traits covary in some species or environments but not in others.

### (c) Associations with individual fitness

Before the early 2000s, studies on individual behavioural traits related to ecological fitness focused on alternative strategies [11,26,42–44]. Early studies focused on the relationship between certain behavioural traits and individual fitness, such as individual aggression predicting response to predators [24,45–50]. Several theoretical/review papers [2,3,51–53] encouraged ecologists to conduct more longitudinal studies to determine if and why behavioural consistency exists in naturalistic settings [5,12,54,55]. A variety of studies demonstrated that individual temperaments predicted ecological outcomes such as survival [4], dominance [56], offspring dispersal [27,57], offspring survival [58], reproductive success [59–61] and anti-predator responses (reviewed in [62]). Further, studies demonstrated that environmental conditions (e.g. predation pressure) can affect the relative numbers of different temperaments in a group, and that the number of different temperaments in a group can influence and be influenced by environmental conditions and can further influence group survival outcomes [26,28,46,63–66]. However, the mechanisms by which these temperaments affect survival, reproduction and overall fitness are not yet clear.

## 3. Pursuing a mechanistic approach

The relative ease of measuring and characterizing temperaments has allowed for an abundance of research on the presence and fitness consequences of behavioural traits in natural populations. To better understand individual consistency of temperament, behavioural syndromes and fitness consequences of temperament we need to pursue an advanced

understanding of physiological mechanisms that underlie temperaments. In the current review, we take advantage of a recent growth in studies that analysed relationship between temperament and physiology in naturalistic conditions (i.e. not under necessarily experimental studies where physiology is manipulated to show cause influences on acute behaviours), and based on these data, we determine whether complex physiological profiles can be identified for different temperaments.

If slight biases in physiological regulation are relatively stable traits within an individual (e.g. [67–74]), then these may support or drive the relative stability, covariance and fitness of behavioural traits [39,70]. To this end, there is already strong evidence that behavioural variance among individuals is systematically associated with neuroendocrine variance (reviewed below) [25,37,48,75]. Several studies have documented *causal* relationships between specific physiological processes and acute displays of certain behaviours. For example, acute injections of central corticotropin-releasing hormone (CRH) and CRH receptor agonist decrease exploratory behaviour in rodents and CRH antagonists increase exploration [76,77]. However, experimental evidence where physiological processes are manipulated in a chronic manner to stimulate trait-like physiological and behavioural regulation are rare and difficult to conduct. Thus, we focus on non-experimental/correlational studies where temperaments are compared to physiology.

Moreover, one-to-one relationships between a behavioural trait and a physiological system do not consider the co-selection of multiple traits nor the complex signalling that occurs among physiological systems. We propose that a more comprehensive understanding of a suite of key physiological processes associated with temperaments is necessary to determine mechanisms that support consistent temperaments, behavioural syndromes, and associated fitness and health consequences. Only by understanding multiple covarying physiological regulatory biases (e.g. elevated sympathetic reactivity, dampened adaptive immune responses, etc.) associated with temperaments can we determine why some temperaments are more fixed, why some temperaments covary and why some temperaments survive better in some environments but not others. With these points in mind, in the current review, we synthesize the research on different physiological mechanisms that have been associated with basic behavioural traits, and determine if specific temperaments are associated with specific physiological profiles (i.e. physiological biases among several systems).

# 4. Connecting temperament to physiology

We focus on correlational studies rather than causal experimental studies where a biological process was manipulated to determine impact on behaviour. We use this focus because correlational studies are more abundant, and they often compare behavioural traits to basal physiological function. Basal physiological regulation is important in that it is more likely to be related to consistent/chronic behavioural patterns and associated fitness outcomes. Finally, we focus on physiological processes that have been frequently studied and that probably influence health and fitness (sympathetic, HPA and immune regulation).

We conducted a literature review on physiological mechanisms associated with the five temperament categories identified by Réale *et al.* [12] and proactive–reactive coping styles, which have been identified as complex behavioural and physiological traits that may be akin to behavioural syndromes [37]. The search terms that we used are in electronic supplementary material, table S1. We removed studies that did not focus on animal biology (i.e. chemistry, physics, etc.), this yielded a list of 14 723 papers. 1702 of these papers were not primary source articles and were removed from the literature search. We then removed papers that did not include at least one of the above-listed temperaments and at least one of the above physiological mechanisms and removed human studies. Based on these criteria, we arrived at a final list of 145 papers.

Results of this search are summarized in table 1. The table shows the number of papers (and total sample size) that indicated a positive, negative or no relationship between each temperament and physiological process. A 'positive' relationship indicates that individuals with a certain temperament showed evidence of an upregulation in the specific physiological process. These relationships were defined based on study results that identified a significant difference ($p < 0.05$) in physiology associated with temperament. Specific information and references from each study that contributed to table 1 are in electronic supplementary material, table S1. To indicate the relative power of relationships between specific temperaments and specific physiological processes, we included total sample size across all studies (in parentheses) in each cell. Electronic supplementary material, table S1 gives detailed information about sample size for each paper.

## (a) Exploration

Exploration, defined as an individual's tendency to engage novel situations, is one of the most common temperaments studied. Highly exploratory individuals tend to disperse more [12,27,78,79] and thus will have more interactions with novel environments and conspecifics compared to less exploratory individuals.

Many studies have documented distinct physiological mechanisms associated with exploration (primarily in rodents and birds). Across species, highly exploratory individuals tend to have elevated sympathetic reactivity (electronic supplementary material, S1–S3; cf. electronic supplementary material, S4, S5) and either dampened or heightened glucocorticoid (GC) responses to stressors [69] (electronic supplementary material, S1, S6, S7, S10, S11, S16, S17, S20, S22, S24–S26, S32, S34, S38–S45; cf. electronic supplementary material, S2, S8, S9, S15, S18, S21, S43, S46–S50), with no relation to basal GC concentrations [69] (electronic supplementary material, S2, S5–S24; cf. electronic supplementary material, S25–S38). Immune function in exploratory individuals tends to favour heightened cell-mediated responses, with minimum energy towards fast-acting non-specific innate immunity, or slower, longer lasting humoral immunity (electronic supplementary material, S17, S24, S25, S31, S44, S51–S56; reviewed in [80–82]; cf. electronic supplementary material, S25, S29, S57). Overall, exploratory individuals have been shown to have heightened sympathetic reactivity and cell-mediated immunity, dampened or heightened HPA reactivity, and lower innate and humoral immunity, with no relation to basal GC circulation.

## (b) Boldness

Boldness, defined as an individual's tendency to engage in risky situations, has been heavily studied in birds and fish.

**Table 1.** Compilation of studies on associations between temperament and physiology. A 'positive' association indicates that *greater* expression of the listed temperament (e.g. exploration) was associated with *increased* activity of a certain physiological process (e.g. sympathetic reactivity), whereas a 'negative' association indicates that *greater* expression of a listed temperament was associated with *decreased* activity of a certain physiological process, and 'none' indicates no relationship between a certain temperament and a certain physiological process. Each cell indicates the number of published papers that showed each temperament-physiology association, and in parentheses the total number of individuals that contributed to all papers.

| temperament | | sympathetic reactivity | basal HPA axis activity | HPA axis reactivity | innate immune response | cell-mediated immune response | humoral immune response |
|---|---|---|---|---|---|---|---|
| exploration | positive | 3 (146) | 7 (545) | 7 (1,633) | 2 (107) | 5 (220) | 1 (10) |
| | negative | — | 7 (462) | 23 (1420) | 4 (294) | 1 (45) | 3 (83) |
| | none | 2 (43) | 22 (2716) | 5 (217) | 2 (73) | 1 (49) | — |
| boldness | positive | 1 (38) | 1 (50) | 6 (274) | 3 (331) | 2 (566) | 1 (28) |
| | negative | 1 (108) | 4 (109) | 6 (289) | 1 (23) | — | 1 (159) |
| | none | 1 (30) | 7 (481) | 9 (3939) | 1 (159) | — | 1 (66) |
| sociability | positive | 1 (16) | 1 (77) | — | — | 1 (7) | 2 (6) |
| | negative | — | 4 (70) | 2 (43) | 1 (7) | — | — |
| | none | 1 (170) | 1 (58) | 3 (279) | — | 1 (26) | — |
| aggression | positive | 1 (170) | 7 (235) | 7 (466) | 3 (126) | 2 (30) | — |
| | negative | — | 4 (259) | 8 (284) | — | — | — |
| | none | — | 1 (30) | 2 (74) | — | — | 2 (612) |
| activity | positive | 2 (108) | 3 (205) | 3 (134) | 1 (23) | — | — |
| | negative | 1 (42) | 1 (125) | 3 (90) | 1 (18) | — | — |
| | none | 1 (132) | 5 (403) | 6 (555) | 1 (27) | — | 1 (592) |
| proactivity | positive | 2 (50) | 1 (147) | 3 (976) | 1 (192) | 1 (218) | 1 (60) |
| | negative | — | 5 (1109) | 9 (1306) | — | 1 (50) | 3 (302) |
| | none | — | 3 (592) | 2 (792) | 2 (130) | — | 1 (80) |

Bold individuals take more risks, expose themselves to novel antigens and obtain more resources than shyer individuals [83] (reviewed in [84]). The degree of an individual's boldness has been associated with learning [85] and tends to predict social dominance within groups [1,37,86].

Physiological processes associated with boldness are less studied than for exploration. The most common physiological process related to boldness is circulating GCs: bolder individuals show either lower or no difference in baseline GCs (electronic supplementary material, S12, S58, S59, S60; cf. electronic supplementary material, S2, S8, S13, S30, S61–S64), and higher HPA reactivity to stressors, particularly in birds (electronic supplementary material, S47, S65–S69; reviewed in [86]; cf. electronic supplementary material, S2, S8, S45, S60, S61, S64, S70), but not in fish (electronic supplementary material, S49, S59, S62, S63, S71, S72; cf. electronic supplementary material, S73, S74). Depending on taxa, bold individuals have elevated (electronic supplementary material, S75), dampened (electronic supplementary material, S76) or no difference in heart rate compared to non-bold individuals (electronic supplementary material, S2). Finally, bolder animals can have higher innate immune reactions (electronic supplementary material, S54, S77, S78; cf. electronic supplementary material, S79, S80). Conflicting results are seen in adaptive immunity, with some studies showing a positive relationship between boldness and cell-mediated immune function (electronic supplementary material, S81 and S82) and others show no clear relationship between boldness and humoral immunity (electronic

supplementary material, S78, S80, S81). Overall, the physiology of boldness is less clear than that of exploration, with a suggestion that boldness is associated with heightened innate and cell-mediated immune function, dampened sympathetic reactivity and humoral immunity, and no clear relationship between boldness and circulating GCs.

## (c) Sociability

Sociability is defined by an individual's tendency to interact with conspecifics—highly sociable individuals interact more frequently with, or with a greater number of, conspecifics. Highly sociable individuals tend to be more central in a group, function as leaders and may affect group dynamics like movement and size [87].

Sociability is frequently studied in primates. More sociable primates tend to have lower basal GCs (electronic supplementary material, S34, S83–S85; cf. electronic supplementary material, S86, S87) and heightened humoral immunity compared to low-sociable individuals (electronic supplementary material, S88–S89; reviewed in [88]). Capitanio [89] and Sloan (electronic supplementary material, S90) concluded that high-sociable primates had a greater cell-mediated immune function but lower innate immune function compared to low-sociable individuals, and only one study contradicts this (electronic supplementary material, S91). The relationship between sociability and HPA reactivity is variable, with two studies showing a negative relationship (electronic supplementary material, S34 and S88) and three

showing no relationship (electronic supplementary material, S50, S91, S92). Last, one study demonstrated that highly sociable goats had higher heart rate reactivity than less sociable goats (electronic supplementary material, S4) while Kralj-Fišer and colleagues (electronic supplementary material, S92) did not observe this relationship in birds. At present, our understanding of the physiological mechanisms that underlie sociability is limited, but there is a suggestion that sociable individuals have either decreased or increased basal HPA activity and have heightened humoral immune function.

### (d) Aggression

Aggressive individuals show more frequent or more intense agonistic reactions towards conspecifics. Aggressive behaviour is species-specific and includes different forms such as territorial aggression, dominance-related aggression and maternal aggression [90]. The aggression trait has been frequently associated with increased exploration and boldness ([91]; [92]), and aggressive individuals tend to locate at the group periphery, be involved in group defence, and have enhanced foraging compared to less aggressive individuals (e.g. [65,93,94]).

There are a limited number of studies on the relationship between aggression and physiological mechanisms in free-ranging animals. The most common measures are related to HPA function: more aggressive individuals, particularly in mammals, have higher basal GCs (electronic supplementary material, S83, S93–S98; cf. electronic supplementary material, S23, S64, S85, S99, S100), whereas the relationship to HPA reactivity is less clear. Equal numbers of studies indicate that more aggressive individuals have dampened (electronic supplementary material, S49, S86, S93–S94, S101–S104; reviewed in electronic supplementary material, S58; [95]) or heightened HPA reactivity (electronic supplementary material, S97, S105–S109) with two that showed no relationship (electronic supplementary material, S64, S110). Studies that showed a negative relationship between aggression and HPA reactivity were often conducted with males and primates while studies that showed a positive relationship included both sexes and a variety of taxa. Aggressive individuals have elevated sympathetic responses compared to less aggressive individuals (electronic supplementary material, S92; reviewed in [41]). Baboons that are more aggressive show faster wound healing [96], lower infection rates [97] and higher lymphocyte numbers suggesting an increased ability to fight off infection (electronic supplementary material, S111). Many studies show that aggressive individuals have heightened innate and cell-mediated immune responses and no relation to humoral immunity compared with less aggressive individuals (electronic supplementary material, S100–S115). Overall, there are few ecological studies of aggressive behaviour, probably because overt aggression is infrequent and therefore difficult to quantify, in well-established social groups in naturalistic settings. Across studies and species, aggressive individuals tend to have greater HPA reactivity, either higher or lower basal circulating GC concentrations, increased innate and cell-mediated immune function, dampened humoral immunity and greater sympathetic reactivity.

### (e) Activity

Activity is a metric of an individual's propensity to move through their landscape. Few studies directly measure activity as a temperament in ecological settings, let alone its connection to physiological mechanisms. Several studies show no

relationship between activity and GC reactivity (electronic supplementary material, S16, S34, S49, S59, S116, S117), and a few studies identified a positive (electronic supplementary material, S48, S118, S119) or negative association with GC reactivity (electronic supplementary material, S45, S61, S120). The same is true for basal GC levels and immunity, with the majority of studies showing no relationship between activity and basal GC levels (electronic supplementary material, S13, S14, S59, S61, S121; cf. S16, S33, S34, S122) or immune function (electronic supplementary material, S54, S115; cf. electronic supplementary material, S31, S79), but a possible positive association with sympathetic reactivity (electronic supplementary material, S76, S121; cf. electronic supplementary material, S116, S123). These studies are evenly split across a variety of taxa. The physiology of this basic temperament requires further elucidation, although a number of studies suggest is not reliably associated with physiology.

### (f) Proactivity

Proactive–reactive coping strategy is defined by coordinated behavioural responses to challenging situations. In particular, proactive individuals are more bold, aggressive and exploratory while reactive individuals are more shy and less aggressive and exploratory. Across studies, proactive–reactive strategies are quantified using different metrics but tend to focus on boldness or aggression. In less dynamic environments, proactive individuals tend to be more abundant while in changing environments reactive individuals tend to have higher fitness [24].

Individuals who are proactive tend to have dampened HPA basal activity and reactivity (electronic supplementary material, S62, S124–S134; reviewed in [24,41]; cf. electronic supplementary material, S129, S135–S137) but heightened sympathetic reactivity compared to reactive individuals (electronic supplementary material, S123, S138; reviewed in [98]). Proactive individuals demonstrate dampened humoral immune responses (electronic supplementary material, S139–S141; cf. electronic supplementary material, S142, S143; reviewed in [55]) and potentially enhanced innate and cell-mediated responses, depending on species (electronic supplementary material, S138, S143–S145).

### (g) Summary

Overall, many studies have identified physiological processes associated with different temperaments. When compiled across species, some regular patterns emerge for certain behavioural traits. In particular, the physiological profile for exploratory, aggressive and proactive individuals is more consistent than the profile for sociable, or active individuals. However, we note that across studies there is a good deal of variation, and that most studies to date only focus on one physiological system, with the majority of studies focused on HPA re/activity.

## 5. Considering multi-system physiological profiles of temperaments

### (a) Why we should consider more complex physiological profiles

If temperament is complemented or supported by a unique regulatory bias in a physiological system (e.g. low HPA

reactivity in highly exploratory individuals), then we should expect these complementary behavioural and physiological phenotypes to be co-selected (electronic supplementary material, S48). For example, high-exploration individuals that tend to expose themselves to more novel environments and antigens will be more likely to survive and reproduce if they have rapid sympathetic responses complemented by low-grade HPA reactivity to allow for a strong cell-mediated immune response. Within this natural selection framework, specific temperaments are probably co-selected along with regulatory biases in multiple physiological systems, leading to complex physiological profiles associated with each temperament. In addition, physiological systems signal to one another and are co-regulated thus we must consider complex multi-system physiological profiles.

## (b) Cross-signalling among physiological systems

Different physiological systems signal to one another, thus, at a functional level, a bias in the regulation of one system can bias activity of another system. For example, chronically elevated GC production can downregulate certain aspects of immune function and neuronal signalling [99,100]. Cross-system signalling occurs among the sympathetic nervous system, the HPA axis, the immune system and certain neurotransmitter systems (e.g. [101–103]). These systems have been well-studied, experimentally manipulated, and implicated in behavioural and physiological responses to the environment, although they have not been studied collectively with respect to temperament. Because physiological regulation involves a variety of feedback and feed-forward mechanisms within and among systems, networks of cross-signalling neurological, endocrine and immune response systems may be relatively difficult to shift, which may explain why some temperaments are highly consistent, or even why some temperament traits regularly covary, and why temperament is associated with fitness in different environments.

## (c) Potential physiological profiles of temperament

Given abundant cross-signalling among physiological systems, and co-selection of traits (behavioural and physiological), it is incumbent on animal behaviourists that study proximate mechanisms to consider more complex physiological profiles. Based on evidence so far, and based on behavioural predispositions of each temperament, we can hypothesize on the structure of physiological profiles for each temperament (see next paragraph and figure 1). A better understanding of the potential physiological profiles will help us understand advantages and disadvantages of specific temperaments in different environmental and social conditions.

Based on the current review, high-exploration is the most highly studied temperament in relation to physiological mechanisms and has been associated with heightened sympathetic activity and cell-mediated immunity, and attenuated innate and humoral immunity. Given an exploratory phenotype that frequently engages novel environments, this physiological profile is probably quite adaptive. Specifically, heightened sympathetic activity with attenuated HPA reactivity allows for rapid, sympathetic-driven responses to dynamic environmental conditions without exposing the exploratory organism to frequent high levels of GCs in circulation, which could dampen immunity. In addition,

heightened sympathetic reactivity would support heightened cell-mediated immunity to maximize a relatively rapid and learned immune response to the frequent novel antigens that exploratory individuals expose themselves to when exploring novel habitats. On the other hand, bold and aggressive individuals that expose themselves to more risky conditions would be better supported with a physiological response that allows for more rapid innate immunity, in addition to cell-mediated reactivity in order to heal wounds in the face of danger and aggressive interactions. In this case, heightened sympathetic activity would support rapid behavioural and immune responses that are required in dangerous situations, and dampened basal GC production would allow a heightened rapid immune response. Finally, we can hypothesize that more sociable individuals will benefit from heightened humoral immunity to respond to frequent re-exposure to antigens passed among social partners. In addition, a strong social network and social group centrality that is often associated with sociability may require less sympathetic and HPA reactivity to respond to novelty or threats. Figure 1 displays hypothesized physiological profiles for each of four temperament types.

If we can test and determine potentially complex physiological profiles for different temperaments, then we are equipped to understand mechanisms that may explain temperament consistency versus flexibility, covariation and fitness in different environmental conditions. For example, a similar physiological profile underlying bold and aggressive temperaments (as depicted above and in figure 1) may explain why boldness and aggression often covary within individuals (i.e. often make up a behavioural syndrome). The relative stability of different physiological profiles could help explain the relative stability of different temperaments. And finally, specific physiological profiles associated with a specific temperament can provide clues as to why certain temperaments thrive in some environmental conditions but not in others. For example, a bold or aggressive physiological profile which includes heightened innate and cell-mediated immunity and sympathetic activation may fair less well in environments that have low nutritional resources, but may do relatively well in environments with heavy predation. Additional understanding of any physiological profiles associated with temperament will provide a better understanding of selection pressures associated with each temperament and may help explain relative behavioural flexibility.

## 6. Future directions

There is limited information on naturally occurring physiological regulatory profiles associated with specific temperaments. Few studies have investigated multiple physiological traits as they relate to any one behavioural trait or temperament [41,104,105]. To understand how temperaments are maintained or change across time and conditions, and to determine long-term consequences of temperaments, we need additional information on the multiple, inter-related physiological systems underlying temperament. To accomplish more complex physiological profiling, we need (i) diverse data that include multiple behavioural and physiological measures, (ii) studies conducted in both the natural

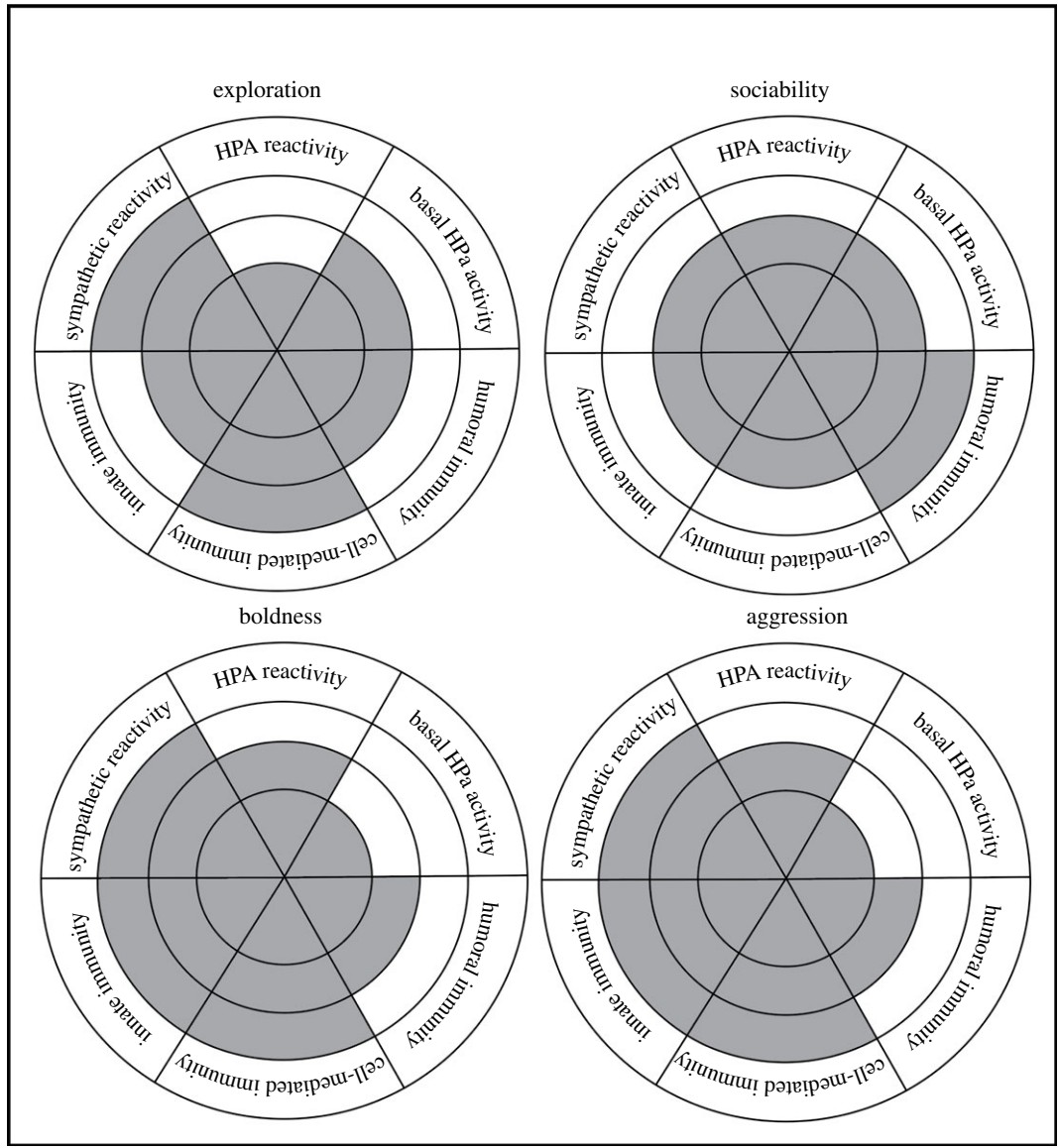

**Figure 1.** Expected physiological profiles for exploration, sociability, boldness and aggression. For each temperament, the grey shading indicates the relative regulation that we might expect of each physiological system (sympathetic reactivity, HPA reactivity, basal HPA activity, innate immunity, cell-mediate immunity and humoral immunity). Grey shading, within any particular physiological system, that extends to the outer-most ring indicates upregulation of that physiological system, while shading that only extends out one ring from the centre indicates downregulation of that system. Shading that reaches the middle ring indicates neither up- nor downregulation of that physiological system. We present hypothetical physiological profiles for the four temperaments that have been most frequently studied.

habitat and controlled laboratory conditions and (iii) a prioritized list of physiological processes that can be quantified in field and laboratory settings.

A starting point to prioritize specific physiological systems to measure is to identify those processes that have an important functional influence on behaviour and fitness. This can build on previous work that has focused heavily on HPA regulation. The neuroendocrine system mediates rapid and long-term behavioural changes and affects multiple physiological systems (reviewed in [106,107]). In particular, given an interest in potential fitness and survival outcomes associated with temperament, a richer understanding of temperament-specific immune regulation biases should be a specific area of increased research (specific immune measures and challenges are reviewed in [105]). In addition, to the neuroendocrine, immune and sympathetic mechanisms that were reviewed here, an easily accessible and highly functional physiological process to include in temperament physiological profiles is gut microbiome diversity. Growing biomedical literature indicates that diversity in gut microbiome impacts temperament-

related physiology, behaviour and health [108,109]. Further, gut microbiome diversity can be measured from fecal samples, making it highly feasible in field research, and this non-invasive metric allows for repeat measurements across time and conditions in free-ranging animals. Metabolic rate is another important physiological mechanism that has been related to behaviour in both endothermic and ectothermic organisms [70,110]. Although metabolic rate is not easily measured in free-ranging animals, prior research on this basic and systemic process suggests that it should be included in temperament-specific physiological profiles, even if limited to laboratory studies.

In addition to identifying other *a priori* physiological processes that are important for fitness, we propose that future work rely on '-omics' genetic expression methods (e.g. transcriptomics and bioinformatics) to identify novel physiological processes related to temperament and that may drive trait flexibility/consistency, covariation and fitness outcomes. Large-scale transcriptomic analyses allow for global estimates of gene expression patterns that are related

to temperament, and with the use of bioinformatics tools, researchers can begin to identify different physiological processes that may be associated with temperament [111,112]. Since temperaments are probably influenced by a multitude of physiological process driven by legions of genes, this work should involve whole-transcriptome or epigenome profiling [86,111,113]. As a first step, the most ideal cells for transcriptomic and bioinformatic analyses may include those present in circulation, given their relative accessibility and systemic function. This system biology approach could be used in both field and laboratory studies.

Overall, a system perspective on the physiological profiles associated with temperament will provide basic information necessary to understand the functional significance of temperament and to understand which temperaments are most suited for specific environmental conditions. This will require cross-disciplinary research with behavioural and physiological experts. Interdisciplinary work will lead to large-scale breakthroughs on how and why individuals systematically behave differently from one another, how these differences are propagated over time, and how they lead to different survival, fitness and health.

**Data accessibility.** This article has no additional data.

**Authors' contributions.** E.K.M.: data curation, visualization, writing the original draft, writing the review and editing; E.Y.: data curation, writing the review and editing; S.A.C.: conceptualization, supervision, writing the original draft, writing the review and editing.

All authors gave final approval for publication and agreed to be held accountable for the work performed therein.

**Competing interests.** None of the authors has any competing interests.

**Funding.** We received no funding for this study.

**Acknowledgements.** This paper is dedicated to Victoria Braithwaite, PhD, who committed her life's work to investigating causes and consequences of animal behaviour and cognition. Her work was creative, groundbreaking, unique, controlled and thoughtful. She dared to ask non-standard questions to truly advance scientific thought and understanding. And she used her science to advance societal goals and to train the next generation of scientists. Victoria provided helpful feedback on initial versions of this paper; she would be pleased to know that it is now published. She is deeply missed (S.A.C.).

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
