## [Peer Review File · Proceedings of the Royal Society B: Biological Sciences]

Review History

RSPB-2021-0802.R0 (Original submission)

Review form: Reviewer 1 (Valeria Bonapersona)

Recommendation

Major revision is needed (please make suggestions in comments)

Scientific importance: Is the manuscript an original and important contribution to its field?

Excellent

General interest: Is the paper of sufficient general interest?

Good

Quality of the paper: Is the overall quality of the paper suitable?

Good

Is the length of the paper justified?

Yes

Should the paper be seen by a specialist statistical reviewer?

No

Do you have any concerns about statistical analyses in this paper? If so, please specify them explicitly in your report.

No

It is a condition of publication that authors make their supporting data, code and materials available - either as supplementary material or hosted in an external repository. Please rate, if applicable, the supporting data on the following criteria.

Is it accessible?

N/A

Is it clear?

N/A

Is it adequate?

N/A

Do you have any ethical concerns with this paper?

No

Comments to the Author

It was a pleasure to read your manuscript. In the attachment, I have provided a general overview, along side major and minor points. I hope that this feedback will help you to improve the manuscript.

Review form: Reviewer 2

Recommendation

Major revision is needed (please make suggestions in comments)

Scientific importance: Is the manuscript an original and important contribution to its field?

Good

General interest: Is the paper of sufficient general interest?

Acceptable

Quality of the paper: Is the overall quality of the paper suitable?

Acceptable

Is the length of the paper justified?

Yes

Should the paper be seen by a specialist statistical reviewer?

No

Do you have any concerns about statistical analyses in this paper? If so, please specify them explicitly in your report.

Yes

It is a condition of publication that authors make their supporting data, code and materials available - either as supplementary material or hosted in an external repository. Please rate, if applicable, the supporting data on the following criteria.

Is it accessible?

N/A

Is it clear?

N/A

Is it adequate?

N/A

Do you have any ethical concerns with this paper?

No

Comments to the Author

The authors present a review of the many relationships between different dimensions of animal personality and different physiological systems. After introducing animal behavioral phenotypes and how they relate to animal personality, they present links to fitness, and then mechanisms. Through this proximate rationale they link animal personality to physiology. Following this introduction they review 6 important personality dimensions – exploration, boldness, sociability, aggression, and activity, as well as proactivity (though it is not one of the dimensions Reale emphasizes). After summarizing these basic, single system-single domain associations they go on to discuss more complex relationships between personality and physiology, and between physiology and physiology, before wrapping up with some suggestions for future research.

The authors have undertaken an unenviable endeavor in seeking to review this literature, but that certainly does not make this an unimportant attempt. This is a fraught research area with many engines of variance and/or noise that have tended to make generalization difficult. Both personality and physiology can be difficult to measure, though in their own ways. However, this review (of mine) is not a critique of the field, though I think it is very important for the authors to consider through their review how we can begin to make sense of this field. I believe the authors have taken some steps in the right direction, but there is room for greater clarity and improvement.

One of the major challenges of reviewing comes from the interaction of two weaknesses. The literature is messy and difficult to summarize, and the impact is unclear. On the second point, we might ask, why is this research area compelling? I do not think the authors have made this case yet. There are rhetorical gaps between the body of evidence, what we as scientists ought to make of it, and how we can go forward.

This review is an important update on the state of the field, or rather, it needs to be an important update, otherwise I am not sure its *raison d'être*, as a review. Overall, I believe the authors need to accurately present the literature along with some content on the *state* of the literature, i.e. controversies/disagreements, because this review needs to present options on how to develop the field from here, and those options need to be justified. The authors should not be shy about pointing out systematic weaknesses, how to fix them, and how to move forward.

One last general comment before those that go line by line, please please please, if it is at all possible, put clear line numbers in the margins of your document.

PG. 5 - “there is current literature on specific behaviors and specific physiological processes” – what does this mean? It is too vague. Please unpack this at least somewhat.

PG. 5 - The focus on correlational studies: this should be mentioned earlier in the manuscript, in the abstract, and perhaps the title.

PG. 9 - Proactivity is first mentioned as a trait on page 5, though not properly introduced until here, page 9. Proactivity is not one of the five dimensions that Reale highlights, though that

review is quite old now. I have no problem with proactivity/reactivity being a focus of the authors, but currently it comes a bit out of nowhere and should be better introduced and integrated. A table describing all of the personality dimensions might be beneficial.

PG.11 – I appreciated the example, though I wonder if the authors might present a diagram to illustrate the example as well. A path diagram would suit, I would think; a directed acyclic graph (DAG) would be best, but that might not be appropriate.

On this, I do also worry that presenting a path diagram might unfairly simplify what are complex relationships between and within systems, and that a DAG might be impossible because these systems and relationships are not acyclic. In that case, how are we to study these systems through a causal lens, or even without one? If the authors have an answer they are confident in, excellent, though I think that some consideration of complexity theory would do this section of the manuscript some good.

Figure 1 – This figure is the main illustration of results, so it needs to be strong. Right now it is confusing and leaves much to be desired.

- The colors are confusing and do not add any extra information. I couldn't tell when colors were darker and lighter – this isn't working.
- The +/- signs are too small, and probably add unnecessary confusion. Either but +, -, or +/- in every sector, or don't use them at all.
- Why only these 4 personality dimensions of the ones the authors discuss? The reasoning should be in the main text or caption somewhere; I didn't find it.
- The authors might need to get rid of the ring. Is there any meaning to one system being closer to another? Moreover, the labels look like they're part of the chart whilst inside the chart.
- Please add a key/legend/scales.
- I suggest the authors try something along these lines: a faceted chart, personality dimensions along the top (x), physiological system along the side (y), and direction of effect, frequency, reliability, etc consistently displayed in each facet.

On the data in this table and “findings” more generally: Though it may be beyond the scope of the review, I would strongly encourage the authors to give us something more quantitative. Since these are correlational studies, we should be able to see the range of correlations for given dimension in a given physiological system, if not the meta-analytic correlation. We ought to be able to see some quantitative measure of heterogeneity as well. Right now the reader has little idea what size of effects we are grappling with, and though this may not be an in-depth review, this is quantitative science and effect sizes provide absolutely bits pieces of information.

PG. 13 – At the end, where genomic methods are introduced, the authors would be remiss to not also mention epigenetics. The epigenome is fast becoming a source for future generations of biomarkers, many of which are closely related to physiology.

Decision letter (RSPB-2021-0802.R0)

28-Apr-2021

Dear Dr Cavigelli:

I am writing to inform you that your manuscript RSPB-2021-0802 entitled "A physiological profile approach to animal personality: Steps to understand the functional significance of behavioural traits" has, in its current form, been rejected for publication in Proceedings B. The referees do think the topic is important, and that the review has potential, but that substantial revisions are

necessary. I agree -- I thought the coverage of the literature was good and the need for better integrated research is unarguable. However, like the referees, I finished the paper feeling opportunities for clarification had been missed and I felt somewhat flat. That said, these problems are potentially fixable. With this in mind we would be willing to consider a resubmission, provided the comments of the referees (and mine, pasted below) are fully addressed. However please note that this is not a provisional acceptance.

The resubmission will be treated as a new manuscript. However, we will approach the same reviewers if they are available and it is deemed appropriate to do so. Please note that resubmissions must be submitted within six months of the date of this email. In exceptional circumstances, extensions may be possible if agreed with the Editorial Office. Manuscripts submitted after this date will be automatically rejected.

- 1) A 'response to referees' document including details of how you have responded to the comments, and the adjustments you have made.
- 2) A clean copy of the manuscript and one with 'tracked changes' indicating your 'response to referees' comments document.
- 3) Line numbers in your main document.
- 4) Please read our data sharing policies to ensure that you meet our requirements <https://royalsociety.org/journals/authors/author-guidelines/#data>.

Sincerely,
Innes Cuthill

Prof. Innes Cuthill
Reviews Editor, Proceedings B
mailto: proceedingsb@royalsociety.org

Editor's comments

I think the terminology needs to be more precise – if that involves criticising the field as a whole, so be it. Starting with the abstract: “individual differences (i.e. animal personality or behavioural phenotypes)”. The “i.e.” could be misleading: individual differences and behavioural phenotype are reasonably interchangeable, but neither are exactly synonymous with animal personality. A behavioural phenotype is used in biomedical circles to refer to the behaviours associated with a (usually genetic pathology), which is certainly not what is meant here. But a phenotype can also consist of one behaviour – for example, just as eye colour is a phenotype, so is the ability to roll one’s tongue – this is not personality. Personality, if it to mean anything useful, is a consistent association of different behaviours, a ‘behavioural syndrome’. This has been one of the major weaknesses in the field – the introduction of a paper talks about animal personality and then, in the Methods, you find that one behaviour is measured – for example, a single measure of latency to do something is labelled ‘boldness’. This is at best unjustified; in my view it’s just plain wrong. Use of “behavioural syndrome” rather than “behavioural phenotype” would be less likely to encourage this sort of mistake. Again, take the abstract: the fact that “current literature describes associations between certain behavioural phenotypes and the activation and regulation of specific physiological systems” is criticised for being too simplistic. But to say that any one behaviour is associated with a specific physiological system is uncontroversial. The assumption that is really being criticised is that a whole suite of superficially unconnected behaviours (a behavioural

syndrome, or personality trait) can be explained by the level of activation of just one physiological system. So, my message is that, where you say near the start of the review, “In this review, we use the terms ‘behavioural trait’ and ‘behavioural phenotype’ – it’s best you don’t! Use behavioural syndrome.

Top of page 3: “Historically, these behavioural differences were thought to comprise random variation, however these differences are now thought to express systematic differences that are trait-like and that promote different survival or life history strategies [5,21,22]” This would be a good place to point out a falsehood in the animal personality literature: that individual differences were ignored until this new wave of research. In fact ‘alternative strategies’, underpinned by game theoretic modelling, has been core to behavioural ecology since the 1970s. The key difference is that old-school ‘alternative strategy’ research always addressed a limited number of phenotypes (usually two) with fitness determined in one context. The (only) thing that made post-2000 animal personality research different and interesting is that now correlations between behaviours deployed in different contexts were recognised. Hence my exasperation at the use of ‘personality’ to describe one behaviour that is consistent within an individual – that’s nothing new, and describing its physiological underpinning is no more than the mechanism question of Tinbergen’s original ‘four whys’ which animal behaviourists have studied for decades.

As referee 1 notes, more information is needed on how the systematic review was conducted. What databases were searched, for example, and covering what period? When I copied the Boolean search string from the Supplementary Table legend into Web-of-Sciences, I got 29,190 hits, so either there’s a typo or other criteria were applied. There also needs to be a definition of how a relationship was defined as Positive, Negative or None (Table 1 and Supplementary Table). Significance at $p < 0.05$ for example? It would be far better to present, and analyse, effect sizes. That something is statistically significant or not is a fairly poor estimate of the magnitude of an effect, as it is so dependent on sample size (see e.g. Nakagawa, S. and Cuthill, I.C. 2007. Effect size, confidence interval and statistical significance: a practical guide for biologists. *Biological Reviews*, 82, 591–605).

Figure 1 needs some work. I wasn’t sure what counts as more, less or intermediate colour – is this the area of colour, or saturation? What counts as darker or lighter? Is it green, or red, or are they equally dark? Are orange, blue and purple equally dark? Is how well studied something is a continuous scale or categorical?

I think the concluding section could do with giving the readers a reality check and more than a plea for more, and more integrated, research. Certainly, experimental manipulation is the only way to determine causality but, given that one of the messages in the review is that physiological systems are interconnected, how does one isolate the contribution of each one? Similarly, how does one estimate the link between a behavioural syndrome and fitness when, by definition, that syndrome is expressed in multiple contexts and, in some cases (e.g. social behaviours) fitness will depend on the social environment (i.e. fitness is frequency dependent). Achieving “more complex physiological profiling” by the steps outlined on page 13, will not be enough.

Reviewer(s)¹ Comments to Author:

Referee: 1

Comments to the Author(s)

It was a pleasure to read your manuscript. In the attachment, I have provided a general overview, along side major and minor points. I hope that this feedback will help you to improve the manuscript.

Referee: 2

Comments to the Author(s)

The authors present a review of the many relationships between different dimensions of animal personality and different physiological systems. After introducing animal behavioral phenotypes

and how they relate to animal personality, they present links to fitness, and then mechanisms. Through this proximate rationale they link animal personality to physiology. Following this introduction they review 6 important personality dimensions – exploration, boldness, sociability, aggression, and activity, as well as proactivity (though it is not one of the dimensions Reale emphasizes). After summarizing these basic, single system-single domain associations they go on to discuss more complex relationships between personality and physiology, and between physiology and physiology, before wrapping up with some suggestions for future research.

The authors have undertaken an unenviable endeavor in seeking to review this literature, but that certainly does not make this an unimportant attempt. This is a fraught research area with many engines of variance and/or noise that have tended to make generalization difficult. Both personality and physiology can be difficult to measure, though in their own ways. However, this review (of mine) is not a critique of the field, though I think it is very important for the authors to consider through their review how we can begin to make sense of this field. I believe the authors have taken some steps in the right direction, but there is room for greater clarity and improvement.

One of the major challenges of reviewing comes from the interaction of two weaknesses. The literature is messy and difficult to summarize, and the impact is unclear. On the second point, we might ask, why is this research area compelling? I do not think the authors have made this case yet. There are rhetorical gaps between the body of evidence, what we as scientists ought to make of it, and how we can go forward.

This review is an important update on the state of the field, or rather, it needs to be an important update, otherwise I am not sure its *raison d'être*, as a review. Overall, I believe the authors need to accurately present the literature along with some content on the *state* of the literature, i.e. controversies/disagreements, because this review needs to present options on how to develop the field from here, and those options need to be justified. The authors should not be shy about pointing out systematic weaknesses, how to fix them, and how to move forward.

One last general comment before those that go line by line, please please please, if it is at all possible, put clear line numbers in the margins of your document.

PG. 5 - “there is current literature on specific behaviors and specific physiological processes” - what does this mean? It is too vague. Please unpack this at least somewhat.

PG. 5 - The focus on correlational studies: this should be mentioned earlier in the manuscript, in the abstract, and perhaps the title.

PG. 9 - Proactivity is first mentioned as a trait on page 5, though not properly introduced until here, page 9. Proactivity is not one of the five dimensions that Reale highlights, though that review is quite old now. I have no problem with proactivity/reactivity being a focus of the authors, but currently it comes a bit out of nowhere and should be better introduced and integrated. A table describing all of the personality dimensions might be beneficial.

PG.11 - I appreciated the example, though I wonder if the authors might present a diagram to illustrate the example as well. A path diagram would suit, I would think; a directed acyclic graph (DAG) would be best, but that might not be appropriate.

On this, I do also worry that presenting a path diagram might unfairly simplify what are complex relationships between and within systems, and that a DAG might be impossible because these systems and relationships are not acyclic. In that case, how are we to study these systems through a causal lens, or even without one? If the authors have an answer they are confident in, excellent, though I think that some consideration of complexity theory would do this section of the manuscript some good.

Figure 1 – This figure is the main illustration of results, so it needs to be strong. Right now it is confusing and leaves much to be desired.

- The colors are confusing and do not add any extra information. I couldn't tell when colors were darker and lighter – this isn't working.
- The +/- signs are too small, and probably add unnecessary confusion. Either but +, -, or +/- in every sector, or don't use them at all.
- Why only these 4 personality dimensions of the ones the authors discuss? The reasoning should be in the main text or caption somewhere; I didn't find it.
- The authors might need to get rid of the ring. Is there any meaning to one system being closer to another? Moreover, the labels look like they're part of the chart whilst inside the chart.
- Please add a key/legend/scales.
- I suggest the authors try something along these lines: a faceted chart, personality dimensions along the top (x), physiological system along the side (y), and direction of effect, frequency, reliability, etc consistently displayed in each facet.

On the data in this table and “findings” more generally: Though it may be beyond the scope of the review, I would strongly encourage the authors to give us something more quantitative. Since these are correlational studies, we should be able to see the range of correlations for given dimension in a given physiological system, if not the meta-analytic correlation. We ought to be able to see some quantitative measure of heterogeneity as well. Right now the reader has little idea what size of effects we are grappling with, and though this may not be an in-depth review, this is quantitative science and effect sizes provide absolutely bits pieces of information.

PG. 13 – At the end, where genomic methods are introduced, the authors would be remiss to not also mention epigenetics. The epigenome is fast becoming a source for future generations of biomarkers, many of which are closely related to physiology.

Author's Response to Decision Letter for (RSPB-2021-0802.R0)

See Appendix A.

RSPB-2021-2379.R0

Review form: Reviewer 2

Recommendation

Accept as is

Scientific importance: Is the manuscript an original and important contribution to its field?

Good

General interest: Is the paper of sufficient general interest?

Excellent

Quality of the paper: Is the overall quality of the paper suitable?

Good

Is the length of the paper justified?

Yes

Should the paper be seen by a specialist statistical reviewer?

No

Do you have any concerns about statistical analyses in this paper? If so, please specify them explicitly in your report.

No

It is a condition of publication that authors make their supporting data, code and materials available - either as supplementary material or hosted in an external repository. Please rate, if applicable, the supporting data on the following criteria.

Is it accessible?

N/A

Is it clear?

N/A

Is it adequate?

N/A

Do you have any ethical concerns with this paper?

No

Comments to the Author

This is an excellent revision. I have no further substantive comments.

Decision letter (RSPB-2021-2379.R0)

24-Nov-2021

Dear Dr Cavigelli

I am pleased to inform you that your manuscript RSPB-2021-2379 entitled "A physiological profile approach to animal temperament to understand evolution and functional significance of individual differences in behaviour" has been accepted for publication in Proceedings B.

The referee whom we sent it to was completely satisfied with your revisions, so many thanks for doing such a good job. Therefore, I invite you to upload the final version of your manuscript. Because the schedule for publication is very tight, it is a condition of publication that you upload the final version of your manuscript within 7 days. If you do not think you will be able to meet this date please let us know.

To upload your manuscript, log into <https://mc.manuscriptcentral.com/prsb> and enter your Author Centre, where you will find your manuscript title listed under "Manuscripts with Decisions." Under "Actions," click on "Create a Revision." Your manuscript number has been appended to denote a revision. You will be unable to make your revisions on the originally submitted version of the manuscript. Instead, revise your manuscript and upload a new version through your Author Centre.

Once again, thank you for submitting your manuscript to Proceedings B and I look forward to seeing your manuscript in print. If you have any questions at all, please do not hesitate to get in touch.

Best wishes,
Innes Cuthill

Professor Innes Cuthill
Reviews Editor, Proceedings B,
mailto: proceedingsb@royalsociety.org

Reviewer(s)' Comments to Author:

Referee: 2
Comments to the Author(s).
This is an excellent revision. I have no further substantive comments.

Decision letter (RSPB-2021-2379.R1)

01-Dec-2021

Dear Dr Cavigelli

I am pleased to inform you that your manuscript entitled "A physiological profile approach to animal temperament to understand the evolution and functional significance of individual differences in behaviour" has been accepted for publication in Proceedings B.

Data Accessibility section

Open Access

Paper charges

Sincerely,
Proceedings B
mailto: proceedingsb@royalsociety.org

Appendix A

RESPONSE TO REFEREES (MS# RSPB-2021-0802 “A physiological profile approach to animal temperament...”)

We appreciate the thoughtful and expert feedback on the original manuscript. Particularly the feedback about stepping back to provide a more nuanced contribution. We addressed this feedback and addressed editor and referee comments in this revised manuscript. Below we outline changes made to the manuscript in response to each editor/referee point. In the FINAL revised Main Document, all additions/changes are noted with yellow highlighting. (We also included an additional Main Document that includes all tracked changes.) **[Location of specific changes to manuscript are provided in brackets.]**

Editor's comments:

The referees do think the topic is important, and that the review has potential, but that substantial revisions are necessary. I agree -- I thought the coverage of the literature was good and the need for better integrated research is unarguable. However, like the referees, I finished the paper feeling opportunities for clarification had been missed and I felt somewhat flat. That said, these problems are potentially fixable. With this in mind we would be willing to consider a resubmission, provided the comments of the referees (and mine, pasted below) are fully addressed. However please note that this is not a provisional acceptance.

Response: We revised the manuscript to provide more clarity and justification of paper goals. In this process, we provided more information on the importance of this topic, why it is novel, and a clearer analysis of studies that were reviewed and included.

Additionally, we integrated the controversies, the need for clarification in the field and what the next steps should be to fill the gaps in this area. **[throughout manuscript]**

I think the terminology needs to be more precise – if that involves criticising the field as a whole, so be it. Starting with the abstract: “individual differences (i.e. animal personality or behavioural phenotypes)”. The “i.e.” could be misleading: individual differences and behavioural phenotype are reasonably interchangeable, but neither are exactly synonymous with animal personality. A behavioural phenotype is used in biomedical circles to refer to the behaviours associated with a (usually genetic pathology), which is certainly not what is meant here. But a phenotype can also consist of one behaviour – for example, just as eye colour is a phenotype, so is the ability to roll one’s tongue – this is not personality. Personality, if it to mean anything useful, is a consistent association of different behaviours, a ‘behavioural syndrome’. This has been one of the major weaknesses in the field – the introduction of a paper talks about animal personality and then, in the Methods, you find that one behaviour is measured – for example, a single measure of latency to do something is labelled ‘boldness’. This is at best unjustified; in my view it’s just plain wrong. Use of “behavioural syndrome” rather than “behavioural phenotype” would be less likely to encourage this sort of mistake. Again, take the abstract: the fact that “current literature describes associations between certain behavioural phenotypes and the activation and regulation of specific physiological systems” is criticised for being too simplistic. But to say that any one behaviour is associated with a specific physiological system is uncontroversial. The assumption that is really being criticised is that a whole suite of superficially unconnected behaviours (a behavioural syndrome, or personality trait) can be explained by the level of activation of just one physiological system. So, my message is that, where you say near the start of the review, “In this review, we use the terms ‘behavioural trait’ and ‘behavioural phenotype” – it’s best you don’t! Use behavioural syndrome.

Response: We appreciate this feedback on terminology. At the beginning of the revised manuscript, we clarified definitions, concepts, and the goal of the current paper. The goal was to determine if temperament traits that are frequently studied in animals are associated with complex physiological profiles. The context for this goal is that these complex physiological profiles (that include neuronal, endocrine, immune, and other function) are an important mechanism to account for temperament flexibility/rigidity, covariance, and fitness. We use the results of the review to advocate for more comprehensive physiological profiling of temperament. The paper is focused on temperament – i.e. broad, within-individual behavioral traits that are consistent across time and conditions (as outlined in Reale et al 2007: exploration, boldness, sociality, activity, and aggression) – rather than on behavioral syndromes (covarying behavioral traits). We reviewed physiological profiles of the reactive-proactive coping strategies because these have been defined as behavioral & physiological syndromes that involve covariation of different temperaments (e.g. aggression, boldness, exploration) and physiology. **[page 2, and page 3 lines 69-73]**

Top of page 3: “Historically, these behavioural differences were thought to comprise random variation, however these differences are now thought to express systematic differences that are trait-like and that promote different survival or life history strategies [5,21,22]” This would be a good place to point out a falsehood in the animal personality literature: that individual differences were ignored until this new wave of research. In fact ‘alternative strategies’, underpinned by game theoretic modelling, has been core to behavioural ecology since the 1970s. The key difference is that old-school ‘alternative strategy’ research always addressed a limited number of phenotypes (usually two) with fitness determined in one context. The (only) thing that made post-2000 animal personality research different and interesting is that now correlations between behaviours deployed in different contexts were recognised. Hence my exasperation at the use of ‘personality’ to describe one behaviour that is consistent within an individual – that’s nothing new, and describing its physiological underpinning is no more than the mechanism question of Tinbergen’s original ‘four whys’ which animal behaviourists have studied for decades.

Response: We appreciate this feedback. We now reference the important and well-established historical focus on alternative strategies. Our focus in the current paper is to test, propose, and advocate for more complex physiological profiling of different temperaments. We do not focus solely on ‘behavioral syndromes’ – i.e. the covariation of different temperaments – but rather on one of Tinbergen’s original questions about mechanisms underlying reliable behavioral traits. But in this case, we advocate for investigations that include more complex physiological profiles of temperament, and argue that this more complex profiling is necessary to understand the evolution and functional significance of temperament. Too often, mechanistic studies focus on just one physiological system, yet we know that these systems do not function in a vacuum, and that multiple physiological systems influence survival and fitness. We added background and references to the history of alternative strategies to the section called “Associations with individual fitness”. **[page 3 lines 64-69, page 5 lines 117-118]**

As referee 1 notes, more information is needed on how the systematic review was conducted. What databases were searched, for example, and covering what period? When I copied the Boolean search string from the Supplementary Table legend into Web-of-Sciences, I got 29,190 hits, so either there’s a typo or other criteria were applied. There also needs to be a definition of how a relationship was defined as Positive, Negative or None (Table 1 and Supplementary

Table). Significance at $p < 0.05$ for example? It would be far better to present, and analyse, effect sizes. That something is statistically significant or not is a fairly poor estimate of the magnitude of an effect, as it is so dependent on sample size (see e.g. Nakagawa, S. and Cuthill, I.C. 2007. *Effect size, confidence interval and statistical significance: a practical guide for biologists. Biological Reviews*, 82, 591–605).

Response: We appreciate the important issue about variable sample sizes across studies that contribute to Table 1. In the original manuscript, we included sample sizes in the Supplementary Table and not in Table 1 because of *Proceedings* manuscript length and general format. Given the above feedback, in the revised manuscript, we include total sample size across all studies in each cell of Table 1. We did not conduct an effect size analysis because the goal was to show what work has been done (and to highlight a heavy focus on HPA axis measures) rather than arrive at an integration of results, given the limited information available. Because there is not enough work yet to identify physiological profiles for different temperaments from a literature review, our goal was to hypothesize about physiological profiles for different temperaments, and how an understanding of these profiles can help explain different phenomena associated with temperament. **[Table 1 on page 10, pages 18-19, and Figure 1 on page 20]**

Article reduction was based on specific criteria, which are now clarified in the revised manuscript, rather than only in the Supplementary Table. As background, many articles were discarded because they did not address specific temperament traits, or they focused on a specific behavior instead of temperament. In addition, the term “activity” resulted in many papers that did not discuss temperament but rather a physiological process. **[page 8 lines 180-185]**

We clarified our definitions of ‘positive’, ‘negative’, and ‘none’ relationships between temperament and physiology in the text and in Table 1 legend. **[page 8 lines 188-191, page 10 Table 1 legend]**

Figure 1 needs some work. I wasn't sure what counts as more, less or intermediate colour – is this the area of colour, or saturation? What counts as darker or lighter? Is it green, or red, or are they equally dark? Are orange, blue and purple equally dark? Is how well studied something is a continuous scale or categorical?

Response: We redesigned the figure to better depict hypothetical relationships between temperament and multiple physiological mechanisms. The figure is now in grey scale and reflects novel text that we added to the manuscript about potential physiological profiles for each of four basic temperaments: exploration, sociability, boldness, and aggression. **[pages 18-19, Figure 1 on page 20]**

I think the concluding section could do with giving the readers a reality check and more than a plea for more, and more integrated, research. Certainly, experimental manipulation is the only way to determine causality but, given that one of the messages in the review is that physiological systems are interconnected, how does one isolate the contribution of each one? Similarly, how does one estimate the link between a behavioural syndrome and fitness when, by definition, that syndrome is expressed in multiple contexts and, in some cases (e.g. social behaviours) fitness will depend on the social environment (i.e. fitness is frequency dependent). Achieving “more complex physiological profiling” by the steps outlined on page 13, will not be

enough.

Response: We revised the conclusion/future directions section to be more specific and targeted about methods and benefits of identifying physiological profiles associated with temperament. In addition, we removed reference to the experimental studies required, as these are premature at this time. [pages 21-22]

Reviewer(s)' Comments to Author:

Referee: 1

It was a pleasure to read your manuscript. In the attachment, I have provided a general overview, along side major and minor points. I hope that this feedback will help you to improve the manuscript.

Overall comment

Thank you for the opportunity to review this interesting paper. I completely agree with the authors that behaviour and its association to physiology is not a simple one-to-one as we tend to approach it in experimental research. In particular, I like the intent of the author to approach the topic from a comprehensive angle, reviewing (possibly) co-transmitted traits (behavioural and physiological) in light of possible evolutionary theories. I have a few comments mainly related to the structure and the purpose of the review, rather than the content. I hope that they can help the Authors to improve the quality of the manuscript.

Response: We appreciate your constructive feedback on the value of the review topic, and on proposed structural changes to clarify the message of the manuscript. Below we indicate specific changes to the revised paper in response to each comment.

Main points

Question 1 – structure.

The authors mention that the aim of the manuscript is to delign the functional significance, evolution and maintenance of behavioural phenotypes. I found this aim a bit vague in the phrasing, as well as in the following structure of the paper. My impression is that the main goal of the paper is to illustrate co-varying traits not merely as biological or physiological, but as combination. In the manuscript, this topic is covered from page 5 onwards (starting with Section “Connecting behavioural phenotypes to physiology”. To me the previous section “Overview of animal behavioural phenotypes” reads more as an introduction to the following part. If I have indeed understood correctly the aim of the paper, I would suggest shortening this part and including it within the introduction. In this way, the aim could be rephrased and the focuses of the article would be clearer to the reader. If I have not understood correctly the aim of the paper, I would recommend the authors to clarify it better in text.

Response: Thank you for this perspective. We shortened the introduction and clarified our objective to focus on how some of the issues and controversies surrounding animal temperament may be resolved with a more complex understanding of the physiology of temperament. [page 2 lines 45-50]

Question 2 – individual differences

This question partially relates to question 1. In text, the authors refer to behavioural phenotypes as of interest when studying individual differences. Yet, part of the literature investigated is at the population level. To me, it just read a bit off-topic. At the same time, I find it a very relevant

topic. For example, I would be really interested in reading your thoughts about how these co-variances of traits would relate to the population base research. For example, from the lesson learned in individual differences, how would you interpret / design / theorize the population based / experimental research? How should one think about the changes in the population after a manipulation while traits are likely cotransmitted and 'biasing' each other? The authors do mention for example that traits are not co-transmitted across different species. This topic is just mentioned, but if indeed individual differences are a main focus on the paper, I think that these questions should be address in more detail.

Response: This would be an interesting perspective to include in the manuscript. Due to the word limit, and that this was not a particular focus for this manuscript, we decided to not delve into population level analyses. Most of the articles in this paper focus on individual differences and so we did not explore these larger issues of how individual temperaments within a group may affect group dynamics and even feedback to affect individual temperaments.

Question 3 - Methodology

The authors mention in page 5 that they conducted a systematic literature search. Systematic literature search is a very specific methodology, yet no details about the methodology are provided. If indeed the authors conducted a systematic literature search, this should be reported according to the PRISMA guidelines. For more details, please see Cochrane reviews (general), CAMARADES or SYRCLE (for animal-related).

Response: Thank you for noting this. We did not use a systematic literature search or meta-analysis methods; we have clarified the methods used to identify and select papers to include in the review. A full description of search terms are included in Supplemental Table 1. **[page 8 lines 179-185]**

Question 4 – Future directions

I fully agree with the authors about a “complexity” approach to identifying co-transmittable traits across domains (physiological / behavioural). However, I am not quite convinced by the methodology. The authors suggest collecting “system-level physiological information to tap into the current genomic methods”. It is not quite clear to me what this means. I find the following paragraph very abstract, and in all honesty, it is very unclear to me how whole-genome profiling could answer this question.

Response: We clarified the methodologies that can be leveraged to identify suites of physiological processes associated with different temperaments. Specifically, we propose the use of transcriptomic analyses, based on well-selected tissue, to identify basal gene up- and down-regulation associated with specific temperaments. Whole transcriptome data collection and analyses with bioinformatics tools is becoming more attainable, and these techniques would be highly advantageous to identify complex physiological profiles associated with temperament with a systems biology approach. And we removed reference to experimental studies in the “Future Directions” section. **[pages 21-22]**

Question 5 - Theory

When investigating the relationship between behaviour and physiology there are two main common approaches. A group of scientists (often biologically trained) tend to see behaviour as something emerging from physiology. A second group of scientists (often psychologically trained) tend to see physiology as something that changes as a consequence of behaviour. Of course, the truth is probably somewhere in the middle. In this paper, the authors mention multiple times to study “causal relationships between behaviour and physiology”, but to me it is unclear how this would happen, in which framework, etc. I find that the review is written intermingling these two approaches. A more theoretical background / framework throughout the review would improve the contextualization and strength of the points made by the manuscript.

Response: Thank you for this perspective and for identifying the confusion that emerges from the cross-disciplinary theoretical backgrounds. We clarified our theoretical stance, which is primarily biological – i.e. behavioral temperaments are thought to emerge from underlying physiological biases. **[throughout, and page 6]**

Minor points

Point 1

In the introduction, the authors write that they make “data-driven” comparisons. In quantitative research, “data-driven” requires data analysis, often without an a priori contextualization. This commentary is not “data-driven” in that sense.

Response: Thank you for catching this. We removed this reference to a ‘data-driven’ analysis. **[page 2]**

Point 2

A recent commentary (<https://doi.org/10.1016/j.biopsycho.2020.10.010>) criticized the approach of “animal personalities” to behavioural research. I wonder how this is reflected in the points made by the authors in this manuscript.

Response: Thank you for including this article in your comments. We reference this paper and expand on inconsistencies in identifying temperaments in animal research. **[page 3 lines 61-67, page 4 lines 86-90]**

Point 3

I did not fully understand Figure 1. Perhaps this could be transformed in a heatmap? For example, the colour could go from blue (negative association) to red (positive association), where white is no association and grey is not investigated

Response: Thank you for this input and suggestions. We decided to create a simpler figure to depict hypothetical physiological profiles for each of four temperaments. **[page 20, Figure 1]**

Referee: 2

The authors present a review of the many relationships between different dimensions of animal personality and different physiological systems. After introducing animal behavioral phenotypes

and how they relate to animal personality, they present links to fitness, and then mechanisms. Through this proximate rationale they link animal personality to physiology. Following this introduction they review 6 important personality dimensions – exploration, boldness, sociability, aggression, and activity, as well as proactivity (though it is not one of the dimensions Reale emphasizes). After summarizing these basic, single system-single domain associations they go on to discuss more complex relationships between personality and physiology, and between physiology and physiology, before wrapping up with some suggestions for future research.

The authors have undertaken an unenviable endeavor in seeking to review this literature, but that certainly does not make this an unimportant attempt. This is a fraught research area with many engines of variance and/or noise that have tended to make generalization difficult. Both personality and physiology can be difficult to measure, though in their own ways. However, this review (of mine) is not a critique of the field, though I think it is very important for the authors to consider through their review how we can begin to make sense of this field. I believe the authors have taken some steps in the right direction, but there is room for greater clarity and improvement.

Response: We appreciate your feedback on the value of the review topic, and on the suggested changes to the manuscript. Overall, the goal of the paper was to advocate for more complex physiological profiling of temperament if we are to understand how and why temperament is related to fitness and survival. We clarified this goal in the Introduction. Below we indicate specific changes to the revised paper based on each specific comments.

One of the major challenges of reviewing comes from the interaction of two weaknesses. The literature is messy and difficult to summarize, and the impact is unclear. On the second point, we might ask, why is this research area compelling? I do not think the authors have made this case yet. There are rhetorical gaps between the body of evidence, what we as scientists ought to make of it, and how we can go forward.

Response: We appreciate this feedback that we did not clearly explain why this work is necessary and compelling. We revised the manuscript to clarify the significance of identifying physiological profiles associated with temperament. **[throughout MS, but in particular in the “Considering Multi-System Physiological Profiles of Temperaments” section, pages 17 & 19].**

*This review is an important update on the state of the field, or rather, it needs to be an important update, otherwise I am not sure its raison d’être, as a review. Overall, I believe the authors need to accurately present the literature along with some content on the *state* of the literature, i.e. controversies/disagreements, because this review needs to present options on how to develop the field from here, and those options need to be justified. The authors should not be shy about pointing out systematic weaknesses, how to fix them, and how to move forward.*

Response: Thank you for this insightful comment. We have expanded on this by discussing the limitation of behavioral assessment of temperament, the limited repeated estimates of trait stability, the use of just one physiological process associated with these temperaments, and how terminology is not clear about consistency across time vs. across condition. We have included this in the sections “Temperaments are individually-consistent” [page 4 lines 86-90, 93-97], “Covariance of temperaments” [page 5 lines 112-115], and “Pursuing a mechanistic approach” [page 6 lines 135-142].

One last general comment before those that go line by line, please please please, if it is at all possible, put clear line numbers in the margins of your document.

Response: Thank you. We added line numbers to the document. [throughout]

PG. 5 - “there is current literature on specific behaviors and specific physiological processes” – what does this mean? It is too vague. Please unpack this at least somewhat.

Response: We agree that this statement was unclear and have removed it. Instead, we discuss how previous studies focus one-to-one relationships between temperament and physiological systems and the pitfalls of this approach. [page 7 lines 160-167]

PG. 5 – The focus on correlational studies: this should be mentioned earlier in the manuscript, in the abstract, and perhaps the title.

Response: We clarified our reasoning to focus on non-experimental (correlational) studies in the section on “Pursuing a mechanistic approach”. We did not add this information to the abstract or title since this was a methodological decision, but we clarified this approach in the “Connecting Temperament to Physiology” section [page 6 lines 137-142, page 7 lines 150-154]

PG. 9 – Proactivity is first mentioned as a trait on page 5, though not properly introduced until here, page 9. Proactivity is not one of the five dimensions that Reale highlights, though that review is quite old now. I have no problem with proactivity/reactivity being a focus of the authors, but currently it comes a bit out of nowhere and should be better introduced and integrated. A table describing all of the personality dimensions might be beneficial.

Response: We include proactivity-reactivity literature because it is a complex behavioral trait that has received more physiological attention than many of the temperament traits listed by Réale et al 2007. We discuss proactive-reactive temperament in the introduction, in the “Covariance of temperaments” section [page 5]. We clarified this section to better represent proactive-reactive traits. While having a table for each temperament dimension would be ideal, unfortunately, we do not have room in this manuscript. However, to clarify each temperament, we give a detailed definition at

the beginning of each temperament within the “Connecting Temperament to Physiology” section.

PG.11 – I appreciated the example, though I wonder if the authors might present a diagram to illustrate the example as well. A path diagram would suit, I would think; a directed acyclic graph (DAG) would be best, but that might not be appropriate. On this, I do also worry that presenting a path diagram might unfairly simplify what are complex relationships between and within systems, and that a DAG might be impossible because these systems and relationships are not acyclic. In that case, how are we to study these systems through a causal lens, or even without one? If the authors have an answer they are confident in, excellent, though I think that some consideration of complexity theory would do this section of the manuscript some good.

Response: In the revised manuscript, we added more hypothetical examples of temperament-specific physiological profiles, revised Figure 1 to represent these theoretical profiles, and then identified some of the adaptive aspects of these profiles for specific temperaments. We removed reference to studying these processes through a causal lens, and rather refer to a system biology approach rather than to complexity theory. In addition, we highlight that we need additional information on physiological profiles before we can address causal relationships by which temperament may influence survival. **[pages 18-20]**

Figure 1 – This figure is the main illustration of results, so it needs to be strong. Right now it is confusing and leaves much to be desired.

- The colors are confusing and do not add any extra information. I couldn't tell when colors were darker and lighter – this isn't working.*
- The +/- signs are too small, and probably add unnecessary confusion. Either but +, -, or +/- in every sector, or don't use them at all.*
- Why only these 4 personality dimensions of the ones the authors discuss? The reasoning should be in the main text or caption somewhere; I didn't find it.*
- The authors might need to get rid of the ring. Is there any meaning to one system being closer to another? Moreover, the labels look like they're part of the chart whilst inside the chart.*
- Please add a key/legend/scales.*
- I suggest the authors try something along these lines: a faceted chart, personality dimensions along the top (x), physiological system along the side (y), and direction of effect, frequency, reliability, etc consistently displayed in each facet.*

Response: We completely revised Figure 1, which depicts hypothetical physiological profiles underlying four different temperaments. The color and +/- signs have been removed from the figure. The caption explains the shading method that is used with each diagram, and also includes information on why only four temperaments are presented. The figure is referenced in the new section of the manuscript entitled “Potential physiological profiles of temperament...”, where we provided additional information about theoretical physiological profiles associated with temperament. For clarity, in Figure 1, the physiological system labels have been moved to the outer most portion. **[pages 18-20]**

On the data in this table and “findings” more generally: Though it may be beyond the scope of the review, I would strongly encourage the authors to give us something more quantitative. Since these are correlational studies, we should be able to see the range of correlations for given dimension in a given physiological system, if not the meta-analytic correlation. We ought to be able to see some quantitative measure of heterogeneity as well. Right now the reader has little idea what size of effects we are grappling with, and though this may not be an in-depth review, this is quantitative science and effect sizes provide absolutely bits pieces of information.

Response: Thank you for this comment and we agree that the table needed more quantitative information. We added total sample sizes to each cell in the table and clarified what we mean by positive, negative or no relationship. Given the relatively limited data available, the goal of Table 1 was to show the amount of work that has been conducted to date relating different physiological processes to temperament, rather than to provide conclusive evidence of different physiological profiles associated with each temperament. Given the limited data, we did not conduct a meta-analysis or effect size analyses. **[page 10]**

PG. 13 – At the end, where genomic methods are introduced, the authors would be remiss to not also mention epigenetics. The epigenome is fast becoming a source for future generations of biomarkers, many of which are closely related to physiology.

Response: We appreciate this suggestion. We added to this section to clarify and to focus on the power of transcriptomic analyses as a first, broad method to identify complex physiological profiles that may be associated with temperaments. We also referenced epigenetic analyses, which provide a specific underlying method by which gene transcription may be altered in different temperaments. **[page 22 lines 449-450]**